# Who's afraid of the big bad boar? Assessing the effect of wild boar presence on the occurrence and activity patterns of other mammals

Êmila Silveira de Oliveira[1☉], Manoel Ludwig da Fontoura Rodrigues[2☉], Magnus Machado Severo[3☉], Tiago Gomes dos Santos[4☉], Carlos Benhur Kasper[5☉]*

1 Laboratório de Biologia de Mamíferos e Aves (LABIMAVE) e Programa de Pós Graduação em Ciências Biológicas, Universidade Federal do Pampa (UNIPAMPA), São Gabriel, Brazil, 2 Departamento de Zoologia, Programa de Pós Graduação em Biologia Animal, Universidade Federal do Rio Grande do Sul (UFRGS), Instituto de Biociências, Universidade Federal do Rio Grande do Sul, Porto Alegre/RS, Brazil, 3 Instituto Chico Mendes de Conservação da Biodiversidade—Ministério do Meio Ambiente (ICMBio/MMA), Mostardas/RS, Brazil, 4 Laboratório de Estudos em Biodiversidade Pampiana (LEBIP), Universidade Federal do Pampa (UNIPAMPA), São Gabriel, Brazil, 5 Laboratório de Biologia de Mamíferos e Aves (LABIMAVE) e Programa de Pós Graduação em Ciências Biológicas, Universidade Federal do Pampa (UNIPAMPA), São Gabriel, Brazil

☉ These authors contributed equally to this work.
* cbkasper@yahoo.com.br

**Data Availability Statement:** All relevant data are within the manuscript and its Supporting Information files.

## Abstract

Wild boar are considered one the world's worst invasive species and linked to biodiversity loss, competition for resources, predation of native species, and habitat modifications. In this study, we use camera traps to evaluate whether the invasive wild boar had an effect on the medium-sized mammal community of a protected area in southern Brazil. Based on photographic records, we evaluated whether the presence and relative abundance of wild boar was associated with a decrease in diversity or change in activity of medium-sized mammals. All comparisons were made between samples where wild boar were present or absent. The records of each camera during a season were considered a sample. The wild boar was the fourth most common species in the study area being present in 7.8% of the photographic records. The species richness of mammals was not negatively affected by the occurrence of wild boar and most common species did not exhibit changes in the daily activity pattern. However, we recorded an increase in the time elapsed between an observation of wild boar and the record of the next species relative to the average latency period observed among other mammalian species. This average latency period was similar to that observed in the case of large predators such as Puma, and its increase could be reflective partly of the avoidance of native species to wild boar. Nevertheless, our results show that the effect of invasive wild boar on the mammal community is not large as expected.

**Funding:** CAPES (Comissão de Aperfeiçoamento de Pessoal de Nível Superior) Brasil. Scholarship grant to ESO. The funders had no role in study design, data collection and analysis, decision to publish, or preparation of the manuscript.

**Competing interests:** NO authors have competing interests.

## Introduction

Biological diversity faces numerous threats globally, including the effect caused by invasive species, which is often irreversible in many cases [1]. After habitat loss, the invasion of alien species is viewed as the second biggest threat to biodiversity [2]. Invasive species tend to show rapid adaptation to new environments and can affect the ecology of native species due to predation, competitive exclusion, and niche drive causing local extinctions in some cases [3].

The wild boar (*Sus scrofa*) is one the top 100 worst invasive species in the world [2]. The original geographical distribution of wild boar encompasses almost all Eurasia, making this one of the most widely distributed mammal species [4]. Currently the wild boar can be found on all continents except for Antarctica [5]. First records in Brazil date from late 1980's and early 1990's [6], with the species now being found in all ecoregions except for Amazonia, which is an invasion speed of 149.6 km$^2$ year [7].

The invasive potential of the wild boar may be related to hybridization with domestic pigs [8]. In tropical countries, hybrid specimens of up to 350 kg have been recorded [9]. In south eastern Brazil, wild adult males have a body mass of 100 to 130 kg but can reach 200 kg or more [10]. Moreover, the large supply of food and the mild climate contribute to more than one parturition event a year, with four to twelve piglets per farrowing [1]. These factors are associated with the virtual absence (or low abundance) of natural predators, allowing exotic wild boar to achieve high densities.

In several countries in the invasive range, different forms of the feral swine have been linked to the decline of biodiversity. It is known that the European wild boar and free-living and feral swine continue to breed and generate hybrids from North America and South America [10–12]. In spite of morphological differences, here we consider the feral forms of *S. scrofa* as the wild boar, because we believe that over generations these animals tend to become ecological equivalents. The effect on animal communities has been observed in invertebrates, amphibians, reptiles, birds and mammals and is related to competition for resources, predation, habitat use and destruction of nests [4]. In addition to these alarming factors, wild boar can be a reservoir of many viral, bacterial, and parasitic diseases that can be transmitted to native fauna, domestic animals, and humans [1, 13–17].

Although wild boar are found throughout the majority of Brazil, Uruguay, and Argentina, information on its effect on biodiversity is limited. Recently in southern Brazil, it was confirmed that these animals decrease the probability of detection and occupation of mammals, negatively affecting the occurrence of some species [18]. Changes in the activities of native species collared peccary (*Pecari tajacu*) and white-lipped peccary (*Tayassu pecari*) due to niche overlap with the wild boar in the Pantanal [19]. Another study carried out in southeast Brazil found that wild boar alters the physical structure of streams, causing silting in these watercourses [20]. The effect of wild boar on degradation of forest vegetation has also been observed [21]. Such effects may cause a decrease in local diversity; however, only few studies have assessed the effects of wild boar on species or groups of animals.

The effects of wild boar on native animals, especially mammals, are still poorly understood. Could the wild boar be affecting the community of medium- and large mammals in preserved environments? Our study aims to evaluate the possible effects of the presence of wild boar in a preserved environment, verifying whether their presence changes the richness and diversity of mammals or is related to possible changes in their activity patterns. Our hypothesis is that the presence of the wild boar would alter composition and change the behaviour of other mammals. Thus, we predict that (1) presence of wild boar will inhibit the occurrence of certain species, lowering richness and diversity indices of native mammals where the wild boar is present or more active, and (2) the pattern of mammalian activity will differ where wild boar are more active.

## Materials and methods

### Study area

The study was conducted in the National Forest (FLONA) of São Francisco de Paula, located in the region of Campos de Cima da Serra, in the northeast of the state of Rio Grande do Sul, coordinates 29˚25′S, 50˚23′O. The climate of this region is classified as temperate; mild summer and rigorous winter, with severe and frequent frosts (Cfb) [22]. The average annual temperature is 10.5˚C (50.9˚F). The FLONA of São Francisco de Paula is a Conservation Unit of sustainable use that has an area of 1,606 hectares (ha). It is formed mainly by Mixed Ombrophilous Forest, also called as the Araucaria Forest (900 ha of native forest), with reforestations of Brazilian pine *Araucaria angustifolia* (Bertol.) Kuntze (390 ha), *Pinus* sp. (229 ha) and small cultivations of *Eucalyptus* sp. (34 ha). This National Forest is considered as a region of "highest priority" for conservation in Brazil according to the Mapping of Priority Areas for the Conservation of the Atlantic Forest [23]. This Conservation Unit presents a rich diversity with at least 29 species of medium-sized mammals (here considered as > 1 kg) recorded [24]. Wild boars were first recorded in this National Forest in April 2007. In the same year, an attempt was made to control the species, without success, due to the difficulty in their capture. Currently, the wild boar appears well-established and adapted to the region as it has been utilising the National Forest across all seasons [25]. The management plan considered the presence of wild boars as one of the main conflicts currently existing in this Conservation Unit [26].

### Data collection

Data collection occurred in five periods of three months: winter (July to September) and spring (October to December) of 2016 and, autumn (April to June), winter (July to September) and spring (October to December) of 2017. The permissions to conduct this study was guaranteed by the licences numbers 47667–1 and 59075–1 of SISBio/ICMBio and by Edenice Brandão Avila de Souza chief manager of the National Forest of São Francisco de Paula.

During each period, five photographic traps (Bushnell model Aggressor) were installed at five points located on trails and roads inside FLONA. These points were located at an average distance of approximately 1.7 km (ranging from 1.5 to 1.9 km) from each other. Considering 1) the small proportion of introduced tree species; 2) the fact that currently, the blocks of Araucaria Plantations are basically identical to native portions of the Araucaria Forests; and 3) the high mobility of medium-sized mammals that probably encompass different microhabitats of the Conservation Unit (CU); we disregard eventual differences in mammal composition between sample points. However, a brief description of the sampled points is important. Point 1 is located near the north border of the CU in a *Pinus* and *Eucalyptus* lot; Point 2 is located at the center/to the west of CU, in the native Araucaria Forest; Point 3 is located in the south of CU in the old Araucaria Plantations parcels; Point 4 is located at the center/to the east in a mix of *Pinus* and Araucaria Plantations plots; and Point 5 is located near the east border in the Araucaria Plantation Plot. This distribution of cameras was set to cover different sectors of the Conservation Unit. Traps were configured to operate 24 hours a day, with a minimum interval of 30 minutes after each record. Due to this time of interval, all records obtained by each camera were considered as independent. No lures or baits were used.

### Data analysis

Initially, we described the occurrence of medium- and large mammals in the National Forest to understand the context of the environment where the wild boar are invading. Data on the number of records were used as an index of relative abundance of each species. This index is

used to analyse the possible effects of wild boar in our study area, as described below. The photographic records of wild boar in FLONA was used to derive ecological data, including information about relative abundance, the proportion of males, females and young and the activity period.

We considered the records of each sampling point (camera trap) on each field trip as a sample unit. As we have five sites sampled at five time points, this corresponds to 25 sample units. Each sample unit was classified by the presence and absence of wild boar in the corresponding period. Next, we compared diversity estimates of native mammals in samples with and without wild boar by using a non-asymptotic approach based on rarefaction (interpolation) and extrapolation curves. Comparisons were made by inspection of curves for a standardized sample size (i.e. mammal abundance) and for a standardized sample coverage (a measure of sample completeness) [27]. For each type of curve, we plotted confidence intervals (95% based on 50 re-samples by bootstrapping), corresponding to three orders of Hill numbers [28]: $q = 0$ (species richness), $q = 1$ (the exponential of Shannon's entropy index), and $q = 2$ (the inverse of Simpson's concentration index). The analysis was based on species abundance data, available from the software iNEXT online [29]. We also use Spearman's coefficient of correlation as an approach to estimate the effect of wild boar abundance on the abundance of other mammal species. These analyses of correlation were made only including species with 10 or more records.

In order to verify possible temporal responses of mammal species to the presence of wild boar, we calculated the average time elapsed between records of native species as well as the time elapsed until the next species is seen after recording a wild boar. The same procedure was adopted for the puma (*Puma concolor*), which is the top predator in this environment. We compared three groups of time periods (considering each camera individually): 1) time elapsed after a record of wild boar and the next mammal species; 2) time elapsed after a record of a puma and the next mammal species; and 3) time between records of two consecutive species (excluding wild boar and puma). To compare the periods between records, a Kruskal–Wallis test was applied, and a comparison of the groups was verified using the Mann–Whitney *U* test. Statistical analyses were performed using PAST software version 2.16 [30] and R [31]. We also compared the latency between the record of wild boars and puma to verify the possibility of the latter following or hunting wild boars, and the latency of wild boar records after a puma record, to infer if they were avoiding a possible predator.

The daily activity of mammals was analysed and described through circular statistical analysis [32]. Using this method we estimated: a) the mean angle ($\mu$), which represents the direction, i.e. the mean time of day during which each species was active; b) the length of the mean vector ($r$), a measure of data concentration around the daily cycles, ranging from 0 (scattered data) to 1 (concentrated data on the same direction), and; c) the circular standard deviation (SD) and confidence intervals (95% and 99%) for $\mu$. We summarized the major results of daily activity using rose diagrams. Records of all common species (more than 10 records) were separated by taxon, and between samples with or without records of wild boar.

The Rayleigh's Uniformity Test was used to calculate the probability of a null hypothesis that the data are uniformly distributed around the circadian cycles ($P > 0.05$) [32]. Thus, a significant result of the Rayleigh test ($P < 0.05$) is associated with longer mean vector and larger Z values, indicating that the data are not uniformly distributed [33]; or there is a concentration of mammal's activity during circadian cycles. Changes in the activity period of mammals in response of the presence of wild boar were evaluated by comparing the patterns of daily activity exhibited in sampled sites without records of wild boar. Thus, we used the Watson–Williams F-test [32], a non-parametric test that compares the lengths of the mean vectors among samples. If the null hypothesis is rejected ($P < 0.05$), this indicates the mammal species present

changes in patterns of daily activity between these conditions. To compare activity patterns, we used only species that were present in more than 10 records in both conditions (with and without wild boar), as a smaller number of records could affect the statistical power. All circular analyses were performed in ORIANA 2.02 [33]. For the Watson–Williams F-test, we selected to incorporate a correction factor, based on the concentration, to correct for possible bias.

## Results

### Diversity and patterns of circannual activity

With a sampling effort of 1191 night traps, 644 photographic records of 21 medium and large mammal species were obtained, including the wild boar (Table 1). The species with the highest number of records were *Cerdocyon thous* (20.2% of the records), *Dasyprocta azarae* (18.2%), and *Leopardus guttulus* (17.7%). The wild boar was the fourth most common species, with 7.8% of the photographic records.

The wild boar were recorded over all sampling periods and were more frequent in the winter months in both years (Fig 1). Solitary males represent 70% of records of this species.

**Table 1. Photographic records of medium and large mammal species in São Francisco de Paula National Forest, Rio Grande do Sul state, Brazil.**

| Taxon | Sampling points | | | | | Total |
|---|---|---|---|---|---|---|
| | **A** | **B** | **C** | **D** | **E** | |
| Didelphimorphia | | | | | | |
| *Didelphis albiventris* | 0 | 1 | 0 | 0 | 0 | **1** |
| Pilosa | | | | | | |
| *Tamandua tetradactyla* | 1 | 0 | 0 | 0 | 2 | **3** |
| Cingulata | | | | | | |
| *Dasypus novemcinctus* | 0 | 0 | 0 | 1 | 1 | **2** |
| *Dasypus* sp. | 0 | 0 | 0 | 0 | 1 | **1** |
| Artiodactyla | | | | | | |
| *Mazama gouazoubira* | 27 | 3 | 3 | 0 | 10 | **43** |
| *Mazama nana* | 0 | 0 | 1 | 0 | 0 | **1** |
| *Pecari tajacu* | 0 | 4 | 2 | 0 | 0 | **6** |
| *Sus scrofa* * | 7 | 11 | 1 | 3 | 28 | **50** |
| Carnivora | | | | | | |
| *Cerdocyon thous* | 79 | 1 | 11 | 11 | 28 | **130** |
| *Lycalopex gymnocercus* | 0 | 0 | 0 | 0 | 1 | **1** |
| *Leopardus pardalis* | 4 | 6 | 1 | 0 | 16 | **27** |
| *Leopardus guttulus* | 30 | 26 | 11 | 23 | 24 | **114** |
| *Leopardus wiedii* | 4 | 7 | 5 | 0 | 3 | **19** |
| *Leopardus* sp. | 4 | 1 | 3 | 3 | 8 | **19** |
| *Puma concolor* | 26 | 3 | 6 | 3 | 5 | **43** |
| *Puma yagouaroundi* | 0 | 1 | 0 | 0 | 1 | **2** |
| *Lontra longicaudis* | 0 | 0 | 0 | 1 | 0 | **1** |
| *Eira barbara* | 0 | 1 | 0 | 0 | 0 | **1** |
| *Galictis cuja* | 0 | 1 | 1 | 1 | 2 | **5** |
| *Nasua nasua* | 0 | 8 | 0 | 2 | 0 | **10** |
| *Procyon cancrivorus* | 17 | 0 | 1 | 5 | 3 | **26** |
| Rodentia | | | | | | |
| *Cuniculus paca* | 0 | 22 | 0 | 0 | 0 | **22** |
| *Dasyprocta azarae* | 0 | 48 | 1 | 9 | 59 | **117** |
| | **199** | **144** | **47** | **62** | **192** | **644** |

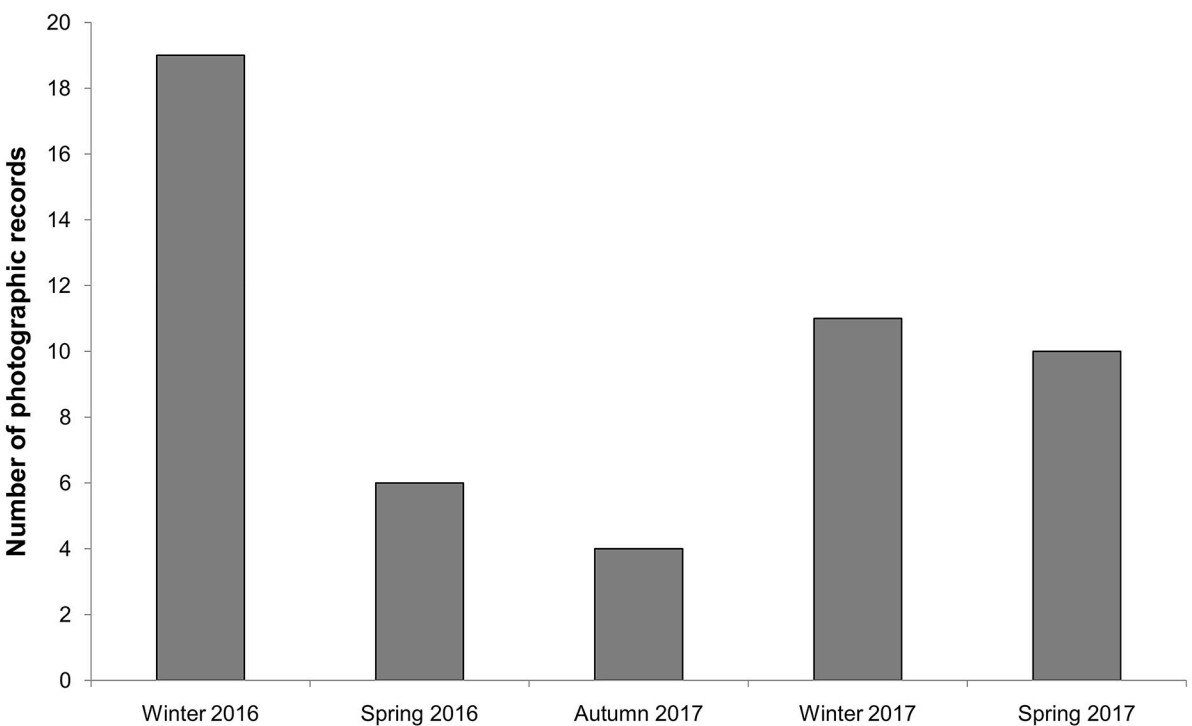

**Fig 1. Number of *Sus scrofa* photographic records along five field sampling periods in São Francisco de Paula National Forest, Southern Brazil.**

Females with farrow accounted for a further 8% of the records, including a video with three females accompanied by 14 farrow. Attempts to identify individuals resulted in the identification of twenty-six individuals of *S. scrofa*. Due to the low recapture rate, it was not possible to estimate wild boar density in the sampled area. Half of the identified individuals occur at the same sample point and almost all (22 individuals) were recorded for only one of three month period. Only three individuals (two females with farrow and one solitary male) were recorded across two months and only one individual (a solitary male) was recorded for three months. None of the identified wild boar was recorded at different sampling points.

When considering the sampling periods (lasting three months) and each sampling point as independent samples, we noted the presence of the wild boar in 60% of the samples. We recorded that samples with and without the occurrence of wild boar do not differ in diversity of medium-sized mammals for order q = 0 (i.e. species richness), as the confidence intervals of both sample-size-based and sample completeness curves overlapped (Fig 2A). However, species diversity for orders q > 0 tended to be higher in samples with the occurrence of wild boar (Fig 2B and 2C). In addition, we did not find any significant negative correlation between the relative abundance of wild boar and the relative abundance of most common species (with more than ten photographic records) (Table 2). The unique significant positive correlation was found between two species: *Leopardus guttulus* and *Dasyprocta azarae*.

## Patterns of daily activity

The average latency period between observing a wild boar and the record of the next species was 53:58 hours. This period was very similar to the latency period observed for the Puma

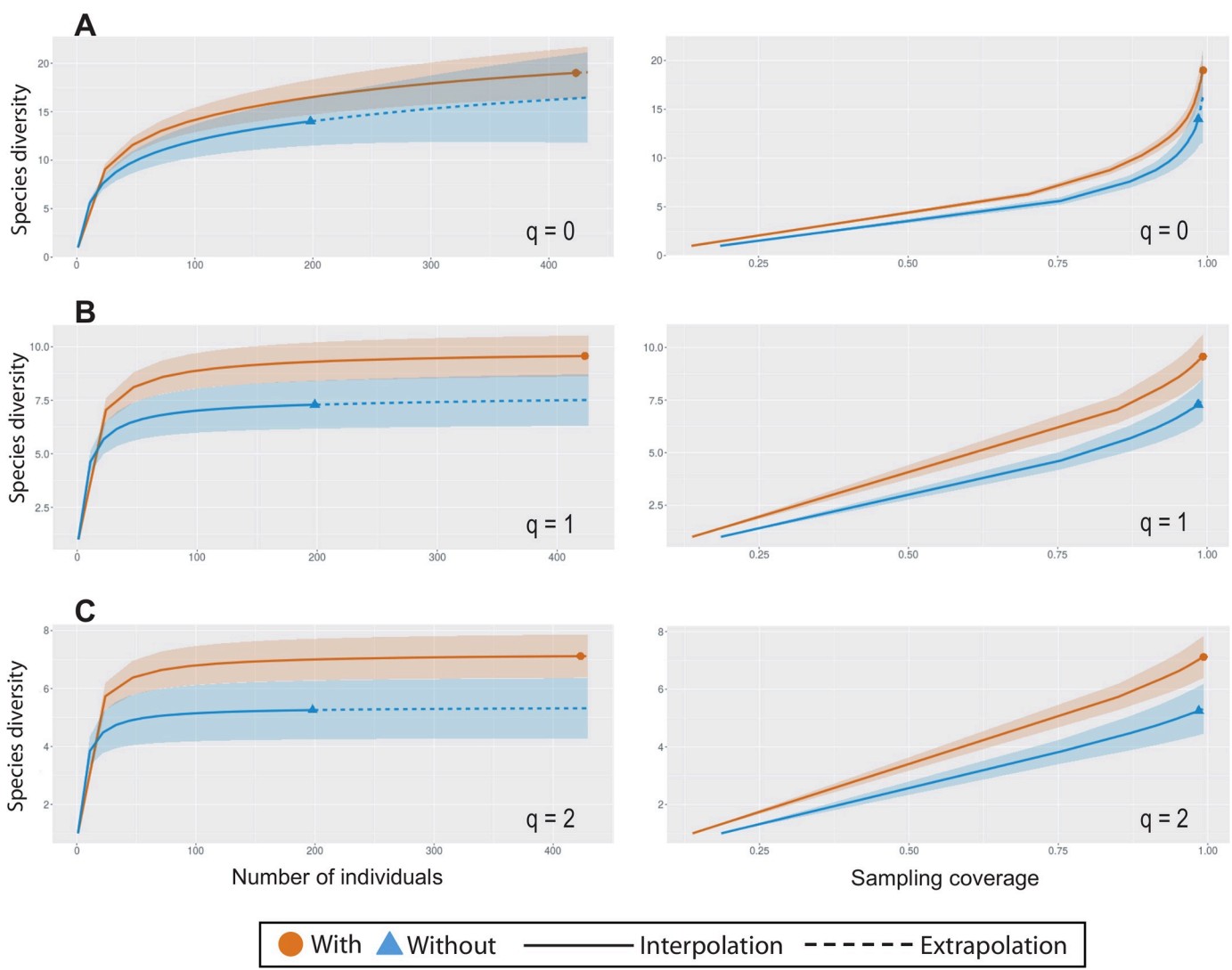

**Fig 2. Comparison of diversity estimates for medium-sized mammals recorded in the samples with and without wild boar in São Francisco de Paula National Forest, Southern Brazil.** Sample-size-based (left) and sample completeness (right) curves for Hill numbers of order q = 0, q = 1, and q = 2. The 95% confidence intervals were obtained by bootstrapping (based on 50 replicates).

(53:39 hours) and differs from that observed amongst other mammalian species, which averaged 41:55 hours. A comparison of latency periods among species showed no difference between the three groups (H = 5.15; p = 0.075). Paired analysis indicates a significant difference only in the latency "after Puma" versus the latency "among other species" (z = 2.13; p = 0.032). The latency of records of puma after the record of a wild boar was more than 8 days (216:25 hours) and the record of wild boar after the record of a puma was 5 days (139:24 hours).

The circular statistical analysis showed that the activity of most mammal species was not uniformly distributed around the daily cycle (Rayleigh test) (Table 3). The highest concentration in daily activity was recorded for *Cerdocyon thous* and *Procyon cancrivorus* (Table 3). Only *Mazama gouazoubira* (in sites without records of wild boar) and *Sus scrofa* presented pattern of activity uniformly distributed around the daily cycle. The wild boar were active during day and night time with peaks of activity at sunrise and first hour of sunset (Fig 3). Females were more diurnal and crepuscular and farrow were recorded exclusively at daytime.

**Table 2. Correlation between the number of records of wild boar and the number of records of the most common species (species with more than 10 records) in São Francisco de Paula National Forest, Southern Brazil.**

| Species | Rho | p-value |
|---|---|---|
| Artiodactyla | | |
| *Mazama gouazoubira* | 0.37 | 0.06 |
| Carnivora | | |
| *Cerdocyon thous* | -0.36 | 0.06 |
| *Puma concolor* | 0.13 | 0.51 |
| *Leopardus pardalis* | 0.20 | 0.32 |
| *Leopardus wiedii* | 0.22 | 0.28 |
| *Leopardus guttulus* | 0.43 | 0.02 |
| *Nasua nasua* | 0.10 | 0.61 |
| *Procyon cancrivorus* | 0.01 | 0.96 |
| Rodentia | | |
| *Cuniculus paca* | 0.28 | 0.17 |
| *Dasyprocta azarae* | -0.44 | 0.02 |

Most of the common species do not exhibit changes in the daily activity pattern when comparing sites were the wild boar was and was not recorded (Table 4). Only *M. gouazoubira* and *P. concolor* changed the daily activity pattern, since the activity of these two species presented higher concentration (i.e. higher mean vector values) in the sites with records of wild boar (Figs 4 and 5).

## Discussion

The National Forrest of São Francisco de Paula is of great importance for biodiversity conservation. We recorded 21 species of medium and large mammals including several threatened

**Table 3. Summary of circular statistical analysis testing for uniformity in daily activity of mammal species monitored in São Francisco de Paula National Forest, Southern Brazil.**

| Species | rc | n | μ | r | Circular SD | 95% CI | 99% CI | Rayleigh Test |
|---|---|---|---|---|---|---|---|---|
| *Mazama gouazoubira* | - | 24 | 13:21 | 0.16 | 110.23˚ | 06:31–20:11 | 04:22–22:20 | 0.59 |
| | + | 21 | 23:14 | 0.50 | 67.07˚ | 21:06–01:22 | 20:26–02:03 | 5.33** |
| *Cerdocyon thous* | - | 68 | 23:39 | 0.60 | 57.83˚ | 22:42–00:37 | 22:24–00:54 | 24.55** |
| | + | 67 | 22:35 | 0.65 | 53.44˚ | 21:43–23:27 | 21:27–23:43 | 28.07** |
| *Leopardus guttulus* | - | 35 | 02:48 | 0.32 | 86.21˚ | 00:06–05:30 | 23:15–06:21 | 3.64* |
| | + | 82 | 02:19 | 0.23 | 98.16˚ | 23:48–04:49 | 23:01–05:36 | 4.35* |
| *Puma concolor* | - | 22 | 20:11 | 0.38 | 79.48˚ | 17:20–23:01 | 16:27–23:55 | 3.21* |
| | + | 23 | 01:42 | 0.46 | 70.92˚ | 23:28–03:57 | 22:46–04:39 | 4.97** |
| *Procyon cancrivorus* | - | 13 | 23:16 | 0.72 | 46.75˚ | 21:22–01:11 | 20:46–01:47 | 6.68** |
| | + | 13 | 00:37 | 0.74 | 44.47˚ | 22:49–02:26 | 22:15–03:00 | 7.12** |
| *Dasyprocta azarae* | - | 12 | 14:39 | 0.55 | 62.68˚ | 11:59–17:19 | 11:08–18:09 | 3.63* |
| | + | 106 | 12:54 | 0.55 | 62.48˚ | 12:03–13:45 | 11:47–14:01 | 32.28** |
| *Sus scrofa* | | 47 | 00:02 | 0.08 | 130.09˚ | 13:53–10:11 | 10:42–13:22 | 0.27 |

Abbreviations: rc = record condition: sites with records of wild boar (+), sites without records of wild boar (-), n = number of observations, μ = mean angle (in time of day), r = mean vector, Circular SD = circular standard deviation, 95% CI = upper and lower confidence intervals of 95% for μ, 99% CI = upper and lower confidence intervals of 99% for μ, Rayleigh Test = Z values and probabilities associated:

*(P<0.05)

**(P<0.01).

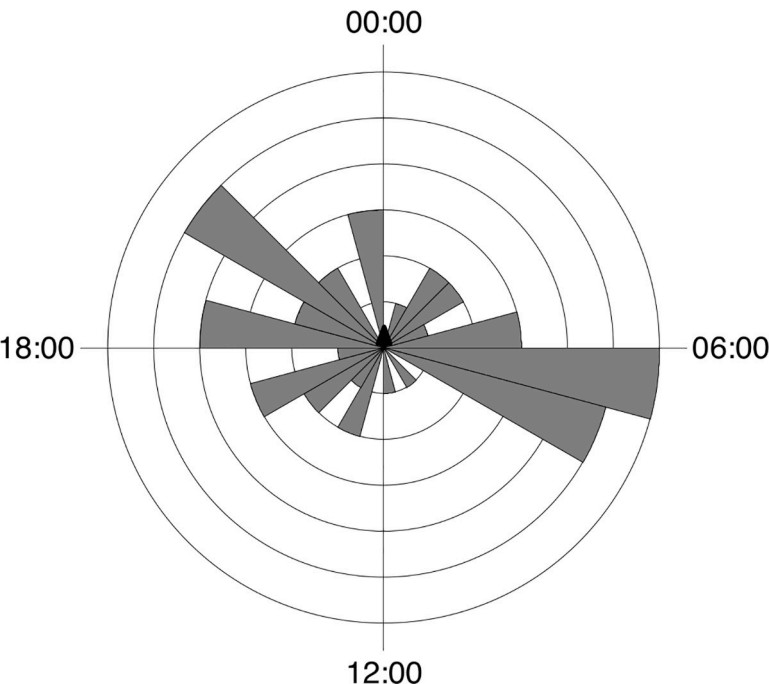

**Fig 3. Activity pattern of *Sus scrofa* in São Francisco de Paula National Forest, Southern Brazil.**

taxa in southern Brazil [34], such as *Tamandua tetradactyla*, *Leopardus guttulus*, *L. pardalis*, *L. wiedii*, *Puma yagouaroundi*, *P. concolor*, *Eira barbara*, *Lontra longicaudis*, *Nasua nasua*, *Cuniculus paca*, *Dasyprocta azarae*, *Mazama gouazoubira* and *M. nana*. Other medium-sized mammals previously recorded at this Conservation Unit have arboreal habitats, such as the Southern Brown Howler Monkey (*Alouatta guariba*), or are typical of the open areas that surround this National Forest and are recorded only occasionally such as the Pampas Deer (*Ozotoceros bezoarticus*), Maned Wolf (*Chrysocyon brachyurus*) or the hog nosed skunk (*Conepatus chinga*) [24, 35].

The higher frequency of wild boar recorded at the National Forest in the winter months may be influenced by the availability of araucaria seeds in this period. Araucaria seeds are part of the wild boar diet throughout its distribution in the Mixed Ombrophilous Forest [10, 21, 36]. The wild boar may affect the regeneration of araucarias trees, considered "critically endangered" by the IUCN [37] not only by the predation of seeds but the extensive disturbance caused when they churn the soil, exposing or damaging the roots [36]. This activity, apart from affecting the regeneration of the species, may also influence the ecology of mammals that

**Table 4. Summary of Watson–Wheeler test comparing daily activity pattern of six most common mammal species in the presence and absence of wild boar, in São Francisco de Paula National Forest, Southern Brazil.**

| Species | W |
|---|---|
| *Mazama gouazoubira* | 7.79 * |
| *Cerdocyon thous* | 2.64 |
| *Leopardus guttulus* | 0.32 |
| *Puma concolor* | 7.52* |
| *Procyon cancrivorus* | 2.39 |
| *Dasyprocta azarae* | 2.29 |

also consume these seeds, especially small rodents [38]. Focused studies are needed to evaluate the effect of wild boar on the population dynamics of the araucaria trees as it was beyond the scope of this study.

The influence of wild boar on the assemblage of medium-sized mammals is difficult to detect and is poorly studied. In our study, samples where wild boar were present exhibited higher diversity of medium-sized mammal orders (q>0), but presented similar diversity estimates for species richness with samples without wild boar (q = 0). The last result is contrary to those obtained by Hegel [18] where samples with the presence of wild boar presented lower mammalian richness. However, our results are congruent with Pantanal, were the wild boar has not been shown to be a direct threat to the native fauna. In this environment, wild boar and other species showed signs of niche division due to morphological and behavioural differences, allowing its coexistence with native pigs, even in periods with low availability of resources [39–41]. We do not believe that wild boar positively affect the environment as the species is still using the same microhabitats selected by other species. In this case, the effect of wild boar can be more severe, as it is occupying preferred habitats of a larger variety of animals. In samples with records of wild boar, we record a decrease of dominance of certain species, such *Cerdocyon thous* that are the most common species in the samples without boar. The correlation of the boar with *Dasyprocta azarae* (the most common species to co-occur with wild boar) may reflect the use of the same environment. Both agouti and wild boar tend to use more forested sample sites when araucaria seeds are more abundant and represent an important resource for the agouti [38, 42]. The correlation of the wild boar with *L. guttulus* may result from a casual consequence of using the same environments.

The wild boar showed a cathemeral pattern of activity. The lack of preference for a certain period of the day, but with peaks of activity at the beginning and the end of the day, are characteristics already expected for the species [42]. The activity distributed throughout the day may indicate a low anthropic effect in the studied area, as wild boar tend to modify their behaviour, altering their use during a certain period in response to anthropic pressure [43]. Another possible influence investigated here were changes in activity pattern of mammals in response to different intensity or frequency of wild boar. The analyses were conducted only with species frequently recorded in samples with and without wild boar. Only two species exhibited some temporal response to the presence of wild boar: the grey brocket deer (*M. gouazoubira*) and the puma (*P. concolor*). The grey brocket deer presented cathemeral activity in samples where wild boar was not recorded and show activity more concentrated at night, especially close to midnight, in samples with the presence of wild boar (Fig 4). Puma that presented peaks in activity in the first hours of night and day in samples with no records of wild boar change to peaks of activity close to midnight (Fig 5). This is interesting as the wild boar exhibits two peaks of activity: near sunset and near sunrise (Fig 3). These changes in the activity of grey brocket deer and puma could reflect some avoidance of wild boar. These findings are counterintuitive because pumas could potentially predate on wild boars. However, previous studies on the food habits of puma in South America indicate that the consumption of wild boars is usually marginal, occurring in sites associated with reduction of natural prey [44–55]. Moreover, the latency observed between records of puma after wild boar records was more than a week and was longer when assessed vice-versa. So, it is possible to suggest that pumas are not actively foraging for wild boars. Pumas probably prey on young wild boars opportunistically, but there are no published data of this behaviour in tropical environments of South America, except by anecdotical information cited by Hegel & Marini (2018) [56].

Although we did not find major shifts in activity pattern of most mammals associated to the presence of the boar, we did observe a delay in the time of recording other mammal species following a record of wild boar. The time gap before other record after the passage of a wild

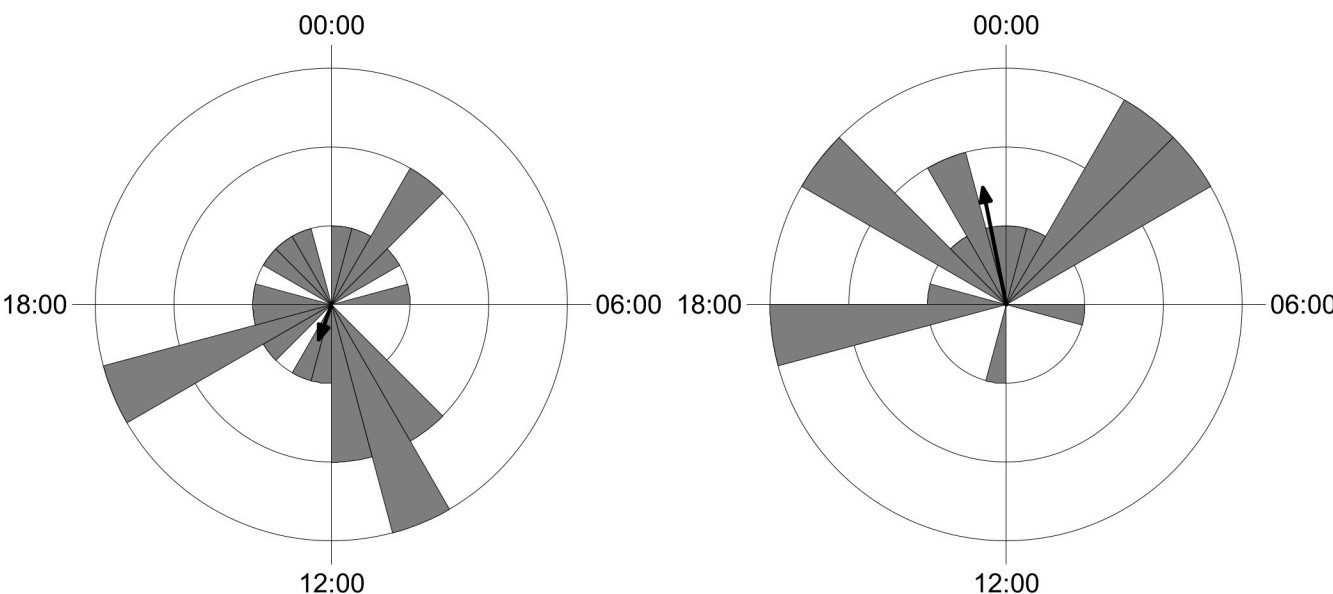

**Fig 4.** Activity pattern of grey brocket deer (*Mazama gouazoubira*) in samples without presence of wild boar (left) and in the presence of wild boar (right).

boar was similar to the time elapsed after the passage of the puma (*P. concolor*). Thus, it is likely the presence of the wild boar has an inhibiting effect on the use of the environment similar to that of a large predator. Therefore, the wild boar could affect the population dynamics of other species (even temporarily) in the use of the habitat.

Biological communities with lower species richness tend to become more prone to biological invasions as they do not offer "biological resistance" against exotic species [57]. Areas with a diverse community of mammals may be more resistant to wild boar invasion due to exploratory competition of other mammals diminishing the invader's resources [44]. In this way, the

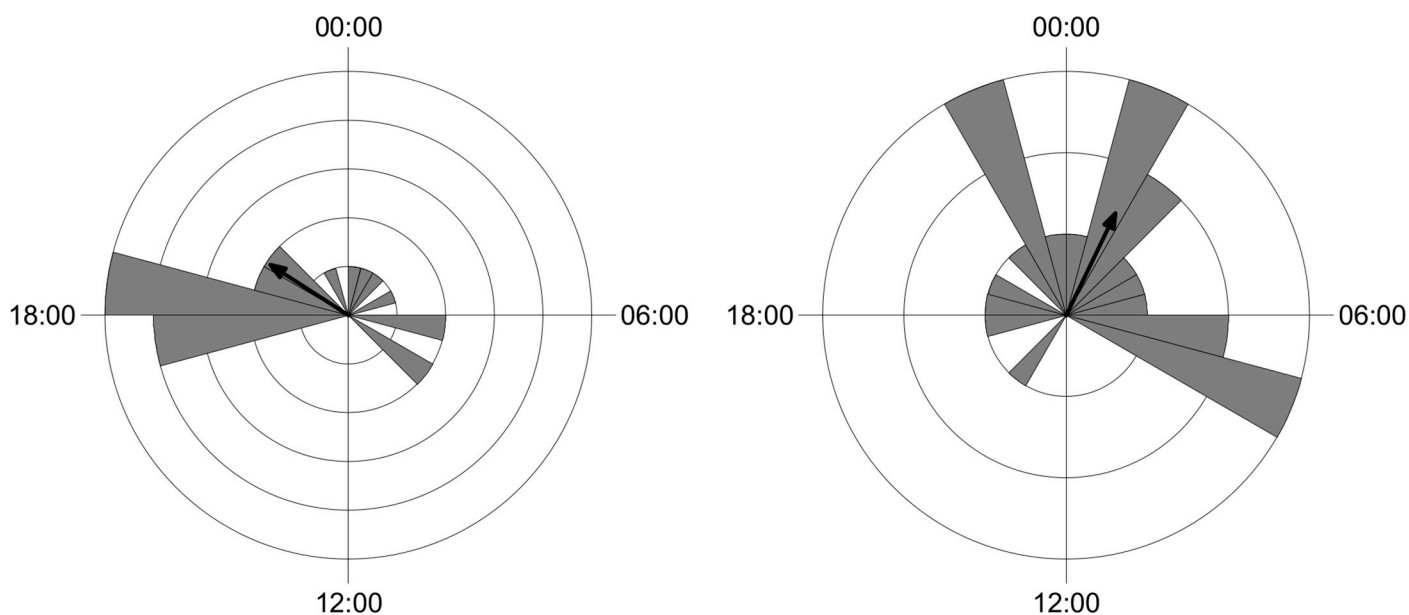

**Fig 5.** Activity pattern of puma (*Puma concolor*) in samples without presence of wild boar (left) and in the presence of wild boar (right).

lack of evidence of the effect of wild boar on the richness and diversity of recorded mammals may suggest that the conservation and high biodiversity of the study area are temporarily 'protecting' native mammal species from the effect. However, it is important to keep in mind that this greater diversity may reduce, but not necessarily prevent, the invasive capacity of the wild boar [44], further reinforcing the importance of the conservation of natural environments and consequently, of native biodiversity. Finally, it is important to understand that effects of invasive species can begin with small modifications in the behaviour of native species, and it is difficult to evaluate what effect this will have on the biological dynamics of the environment [58, 59].

## Supporting information

**S1 Dataset. Datasheet with activity patterns of wild boar and other mammals in São Francisco de Paula National Forest, Rio Grande do Sul state, Brazil, and comparative statistics.** (XLSX)

## Acknowledgments

We thank to São Francisco de Paula National Forest administration for the logistic support, especially to Edenice Brandão Avila de Souza.

## Author Contributions

**Formal analysis:** Tiago Gomes dos Santos, Carlos Benhur Kasper.

**Investigation:** Êmila Silveira de Oliveira, Manoel Ludwig da Fontoura Rodrigues, Magnus Machado Severo, Carlos Benhur Kasper.

**Writing – original draft:** Êmila Silveira de Oliveira, Manoel Ludwig da Fontoura Rodrigues, Magnus Machado Severo, Tiago Gomes dos Santos, Carlos Benhur Kasper.

**Writing – review & editing:** Êmila Silveira de Oliveira, Manoel Ludwig da Fontoura Rodrigues, Magnus Machado Severo, Tiago Gomes dos Santos, Carlos Benhur Kasper.

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
