## [Decision Letter · Decision Letter 0]

27 Nov 2019

PONE-D-19-25832

Who’s afraid of the big bad boar? Assessing the effect of wild boar presence on the occurrence and activity patterns of other mammals

PLOS ONE

Dear Dr. Kasper,

Thank you for submitting your manuscript to PLOS ONE. After careful consideration, we feel that it has merit but does not fully meet PLOS ONE’s publication criteria as it currently stands. Therefore, we invite you to submit a revised version of the manuscript that addresses the points raised during the review process.

While we have only received comments from one reviewer- the comments are very considered and need to be fully addressed

We would appreciate receiving your revised manuscript by Jan 11 2020 11:59PM. To enhance the reproducibility of your results, we recommend that if applicable you deposit your laboratory protocols in protocols.io, where a protocol can be assigned its own identifier (DOI) such that it can be cited independently in the future. For instructions see: http://journals.plos.org/plosone/s/submission-guidelines#loc-laboratory-protocols

We look forward to receiving your revised manuscript.

Kind regards,

Judi Hewitt

Academic Editor

PLOS ONE

Reviewers' comments:

Reviewer's Responses to Questions

**Comments to the Author**

1. Is the manuscript technically sound, and do the data support the conclusions?

Reviewer #1: Partly

2. Has the statistical analysis been performed appropriately and rigorously? 

Reviewer #1: Yes

3. Have the authors made all data underlying the findings in their manuscript fully available?

Reviewer #1: Yes

4. Is the manuscript presented in an intelligible fashion and written in standard English?

Reviewer #1: Yes

5. Review Comments to the Author

Reviewer #1: I find this paper very interesting as it provides insights into a a scarcely investigated mammal community and on a widely distributed problem i.e. the impact of alien species on local animal communities. The area selected is interesting as it hosts a quite rich mammal community and gives the possibility to investigate a complex ecological web. The effort done in collecting data and in the analyses done is appreciable however I have some concerns on the lack of proper description and evaluation of the ecological characteristics of the sampled areas. I think that a sound effort to describe and quantify quantify ecological factors in the five spots might have been very useful to provide better interpretation of the results. I have explained my concerns and given my suggestion in the annotated text I attach to this comment.

6. PLOS authors have the option to publish the peer review history of their article (what does this mean?). If published, this will include your full peer review and any attached files.

Reviewer #1: Yes: Marco Apollonio

---

## [Author Response · Author response to Decision Letter 0]

21 May 2020

We are thankful for the reviewer comments that improve que quality of our manuscript. We respond to each of the reviewer comments for the manuscript in the rebuttal letter.

---

## [Editor Report · Decision Letter 1]

15 Jun 2020

Who’s afraid of the big bad boar? Assessing the effect of wild boar presence on the occurrence and activity patterns of other mammals

PONE-D-19-25832R1

Dear Dr. Kasper,

We’re pleased to inform you that your manuscript has been judged scientifically suitable for publication and will be formally accepted for publication once it meets all outstanding technical requirements.

Kind regards,

Judi Hewitt

Academic Editor

PLOS ONE
---

## [Editor Report · Acceptance letter]

22 Jun 2020

PONE-D-19-25832R1 

Who’s afraid of the big bad boar? Assessing the effect of wild boar presence on the occurrence and activity patterns of other mammals 

Dear Dr. Kasper:

I'm pleased to inform you that your manuscript has been deemed suitable for publication in PLOS ONE. Congratulations! Your manuscript is now with our production department. 

Kind regards, 

on behalf of

Dr. Judi Hewitt 

Academic Editor

PLOS ONE